# Dynamic Mechanical Properties of Rolled Thin-Walled Steel Plates (TWSPs) Used for W-Beam Guardrails under Low and Medium Strain Rates

**DOI:** 10.3390/ma15196504

**Published:** 2022-09-20

**Authors:** Fangfang Liu, Xiaowei Cheng, Yi Li, Manjuan Yang, Yujing Zhou

**Affiliations:** 1Beijing Key Laboratory of Earthquake Engineering and Structural Retrofit, Beijing University of Technology, Beijing 100124, China; 2Key Laboratory of Road and Traffic Engineering of Ministry of Education, Tongji University, Shanghai 201804, China; 3Research Institute of Highway of Ministry of Transport, Beijing 100088, China

**Keywords:** rolled thin-walled steel plate, mechanical properties, low and medium strain-rate region, cold rolling treatment, Cowper–Symonds model

## Abstract

Accurately considering the dynamic mechanical properties of rolled thin-walled steel plates (TWSPs) under low and medium strain rates is the basis of numerical simulations of W-beam guardrails subjected to vehicle impact. Uniaxial tensile tests were conducted on specimens extracted from different locations (flat TWSPs without cold rolling treatment, and the cross-sectional centers and slopes of rolled TWSPs) and under different strain rates ($\dot{\varepsilon }$ = 0.00025, 0.01, and 50 s^−1^). Based on experimental and numerical results, the cross-sectional center of a rolled TWSP is recommended as the representative sampling location for uniaxial tensile tests. Additional uniaxial tensile tests with wider strain rates of 10, 100, and 200 s^−1^ were also conducted on specimens at the recommended sampling location (cross-sectional center) of rolled TWSPs. It was found that the Cowper–Symonds model with parameters of *C* = 40 s^−1^ and *p* = 5 recommend by Symonds significantly overestimated the strain rate effects of the rolled TWSP material in the low and medium strain-rate region. The model with calibrated parameters of *C* = 4814 s^−1^ and *p* = 2.9 was recommended for considering the influences of strain rate effects on the dynamic mechanical properties of rolled TWSP at low to medium strain rates.

## 1. Introduction

W-beam guardrails made by rolled thin-walled steel plates (TWSPs) are important safety facilities on the highway to mitigate harmful consequences of traffic accidents [1]. The crashworthiness of such guardrails under vehicle impact is commonly studied by numerical simulation methods [2,3]. In the numerical analysis, a material constitutive model of rolled TWSP that can accurately represent the yielding, strengthening, failure and strain rate effects is critical for reasonable simulation results. Calle et al. [4] found that ignoring the strain rate effect of steel in numerical simulations of impact scenarios underestimated the impact force of T cross-section beams and overestimated their mid-span deformation. Heo and Kunnath [5] found that when numerical simulations adopted simplified bilinear stress–strain models for steel, the simulated dynamic responses of structural members deviated significantly from the ones measured in the test. When TWSPs were applied in W-beam guardrails, flat TWSPs were often cold-rolled into rolled TWSPs, which meant the mechanical properties of the latter were distinct from those of the former. Although studies have been conducted on the flat TWSPs in the past research [6], limited studies have been conducted on the mechanical properties of rolled TWSPs under different strain rates. Therefore, the material constitutive model of flat TWSPs was commonly used for W-beam guardrails [7,8], which probably resulted in inaccurate predictions for the impact-resistant performance of W-beam guardrails.

During the cold-rolling process of rolled TWSPs used for W-beam guardrails, nonuniform residual stress and strain developed along the cross-sections of rolled TWSPs, which essentially resulted in different mechanical properties for the rolled TWSP material at the various sampling locations. However, few studies have been conducted to evaluate the differences between mechanical properties for the rolled TWSP material at the various sampling locations. Therefore, although detailed requirements about the sampling location of structural steel have been specified in GB/T 2975-2018 (China code) [9] and E8/E8M-16a (U. S. code) [10], no related requirements were specified for the representative sampling location of a rolled TWSP to measure its mechanical properties [11,12,13].

Under vehicle impact, rolled TWSPs in W-beam guardrails have often exhibited low to medium strain rates ranging from 0.0025 to 200 s^−1^ [14]. Due to a lack of testing data for the rolled TWSP material, the Cowper–Symonds model with parameters of *C* = 40 s^−1^ and *p* = 5 is often used to consider the influence of strain rate on its mechanical properties [15,16]. Based on past research, multiple factors, including the chemical compositions and physical processing procedures, have significant effects on the Cowper–Symonds model parameters [17], which has resulted in these parameters varying significantly among various kinds of structural steel. *C* ranges from 0.76 to 230,000 s^−1^, and *p* ranges from 2.0 to 7.3 [18]. Therefore, the Cowper–Symonds model parameters that are suitable for rolled TWSP material need to be further calibrated to accurately capture the influences of strain rate effects on the mechanical properties of rolled TWSPs under impact scenarios.

In this study, static, quasi-static and dynamic uniaxial tensile tests with strain rates of 0.00025, 0.01, and 50 s^−1^, respectively, were firstly conducted on specimens extracted from a flat TWSP (without cold rolling treatment) and the cross-sectional center and slope of a rolled TWSP (with cold rolling treatment). The influences of cold rolling treatment on the mechanical properties of TWSPs were investigated. The cross-sectional center of rolled TWSPs was recommended as a representative sampling location on the basis of the test results and numerical analysis of rolled TWSPs. Based on the representative sampling location, additional dynamic uniaxial tensile tests with wider strain rates of 10, 100, and 200 s^−1^ were conducted to study the influences of strain rates on the mechanical properties of the rolled TWSP material. Finally, calibrated Cowper–Symonds model parameters of *C* = 4814 s^−1^ and *p* = 2.9 were proposed to ensure the Cowper–Symonds model could reasonably reflect the dynamic mechanical properties of the rolled TWSPs. Overall, the novelties and contributions of this study include: (1) A representative sampling location was recommended for uniaxial tensile tests of rolled TWSPs; (2) Cowper–Symonds model parameters were calibrated to reasonably capture the dynamic responses of rolled TWSPs under low to medium strain rates, which was not studied previously.

## 2. Experimental Program

### 2.1. Test Setup and Measurement Devices

In accordance with the regulations in the Health and Safety Executive (HSE) technical report [19] (U.S. code), GB/T 30069.1-2013 (China code) [20] and GB/T 30069.2-2016 (China code) [21], for strain rates used in uniaxial tensile tests, MTS, Zwick/Roell Z100 and Instron VHS 160/100-20 tensile testing machines were used to perform static, quasi-static and dynamic uniaxial tensile tests, respectively, as presented in Figure 1 and Table 1. In the static and quasi-static tests, the predetermined strain rates were set to 0.00025 and 0.01 s^−1^, respectively. The specimens were directly used at the predetermined strain rates. In the dynamic tests, the predetermined strain rates were set to 10, 50, 100, and 200 s^−1^. A set of clamping devices was fabricated to achieve the predetermined strain rates, which included an extension plate, two splice plates, and friction-type high-strength bolts, as depicted in Figure 1d. Similar clamping devices were used in the previous study using the same Instron VHS 160/100-20 tensile testing machine in order to get the dynamic mechanical properties of flax fiber reinforced polymer [22] and polyvinyl butyral [23]. Prior to commencing the test, the specimen was connected to the extension plate by the bolts and splice plates. The extension plate and specimen were connected to the moving jaw and fix grip head, respectively, of the VHS 160/100-20 tensile testing machine. At the beginning of the dynamic test, the moving jaw was accelerated in the tension direction of the specimen to reach the predetermined strain rate. As soon as the predetermined strain rate was reached, the extension plate was clamped by the moving jaw, and the specimen was stretched to failure at the strain rate. The test procedures of static, quasi-static and dynamic uniaxial tensile tests on the TWSP material were regulated in GB/T 228.1-2010 [24] and GB/T 30069.2-2016 [21].

For the static and quasi-static tests, the strain of the specimens was measured using a normal extensometer with a 25 mm gauge length and a fully automated extensometer with a 20 mm gauge length, as shown in Figure 1a,b. For the dynamic tests, the strain of the specimens was obtained from the deformation data recorded by a high-speed camera with a frame rate of 60,000 fps, as shown in Figure 1c. It needs to be noted that room temperature during all the tests was around 25 ± 5 °C.

### 2.2. Specimen Design

The rolled TWSP (Grade DB01 from Jiangsu Huaxia Corporation, Nanjing, China) recommended in the specification for W-beam guardrails (GB/T 31439.1-2015) [11] was taken as the research object. Figure 2 presents the position of the rolled TWSP in W-beam guardrails, cross-sectional dimensions of the rolled TWSP, and the sampling locations of test specimens on it. As illustrated in Figure 2b, the rolled TWSP was 310 mm in width, 85 mm in height, and 4 mm in thickness. The specimens were extracted from the cross-sectional center (LC specimens) and slope (LS specimens) of the rolled TWSP along longitudinal direction and the unrolled flat TWSP (LU specimens) to investigate the effects of cold rolling treatment. It needs to be noted that the flat TWSP and rolled TWSP refer to the TWSP material before and after cold rolling treatment, respectively. All test specimens were processed using wire cutting to satisfy the geometry requirements in GB/T 228.1-2010 [24] and GB/T 30069.2-2016 [21]. The geometric dimensions of specimens in the static, quasi-static, and dynamic tests are presented in Figure 3. In order to match tensile testing machines used for different strain rates, the geometric dimensions of specimens in quasi-static and dynamic tests were different from those of specimens in the static test. The experimental research on the rate-dependent constitutive model of Q420 steel [25] and dynamic tensile behavior of 400 MPa-grade anti-earthquake hot-rolled ribbed bar [26] also adopted different geometric dimensions for specimens in different strain rates.

## 3. Effects of Cold Rolling Treatment on Mechanical Properties of TWSPs

### 3.1. Test Specimens

Twenty-seven specimens extracted from different locations of TWSPs were stretched to failure to investigate the effects of cold rolling treatment on the mechanical properties of the TWSP material, as listed in Table 2. LS and LC specimens were extracted from the cross-sectional slope and center of the rolled TWSP, respectively, which was clarified in Section 2.2. LU specimens were extracted from the unrolled flat TWSP. The predetermined strain rates were set to 0.00025, 0.01, and 50 s^−1^, for the static (S), quasi-static (QS), and dynamic (D50) tests, respectively. To ensure repeatability and enhance the reliability of the test data, each group of tests had three identical specimens, and the average values of the test results were studied.

### 3.2. Test Results and Analysis

#### 3.2.1. Tests Results

Figure 4 depicts the relationship between the measured engineering stress *σ*_eng_ and engineering strain *ε*_eng_. The measured engineering stress was the average stress of three specimens at identical strain. The photographs of LS specimens after uniaxial tensile tests are also given in Figure 4 to illustrate the failure modes of all specimens under the three strain rates. All test specimens exhibited obvious plastic necking deformation during uniaxial tensile tests. The fractures of the specimens were fibrous and dark gray by visual examination, indicating that ductile failure occurred in the strain rate range of 0.00025 to 50 s^−1^ [27]. In both static and quasi-static tests, obvious yield plateaus were observed in engineering stress–strain curves for LU specimens, and the yield plateau disappeared for LS and LC specimens. In the dynamic tests, the yield plateau for LU specimens was no longer apparent, indicating that the increase in strain rate decreased the yield plateau.

For the measured engineering stress–strain curves with or without yield plateau, the strength parameters (i.e., yield strength *f*_y_, tensile strength *f*_u_ and fracture strength *f*_t_) and deformation parameters (i.e., modulus of elasticity *E*, total percentage extension at maximum force *A*_gt_ and percentage elongation after fracture *A*) are defined in Figure 5. The yield strength of the steel without yield plateau was taken as the stress corresponding to plastic strain of 0.2% [28]. The calculated strength and deformation parameters for all specimens are listed in detail in Table 3. Notably, engineering stress and engineering strain were used for the analysis of test results (Section 3 and Section 5.1), as defined in design codes [10,24]. True stress *σ*_true_ and true strain *ε*_true_ were used in the calibration of Cowper–Symonds model parameters (Section 5.2 and Section 5.3), and equivalent stress *σ*_eff_—equivalent plastic strain *ε*_eff_ curves were used in numerical simulations (Section 4 and Section 6) to satisfy the requirements of the finite element software and theoretical model [4,18,29]. The translation relationships between engineering stress *σ*_eng_ and engineering strain *ε*_eng_; true stress *σ*_true_ and true strain *ε*_true_; and equivalent stress *σ*_eff_ and equivalent plastic strain *ε*_eff_, are listed in Formulas (1)–(4).


*σ*_true_ = *σ*_eng_(1 + *ε*_eng_)(1)
*ε*_true_ = ln(1 + *ε*_eng_)(2)
*σ*_eff_ = *σ*_true_(3)
*ε*_eff_ = *ε*_true_ – *f*_y,ture_/*E*(4)


#### 3.2.2. Analysis of Strength Parameters

Figure 6 presents the relationship between the yield strength *f*_y_ and tensile strength *f*_u_ of the TWSP material at three sampling locations and strain rates. The following observations could be made: (1) The static, quasi-static, and dynamic yield strengths of LS specimens were 10%, 7%, and 15% lower than those of LU specimens without cold rolling treatment, respectively, as shown in Figure 6a. The static and quasi-static yield strengths of LC specimens were 4% and 6% higher than those of LU specimens, respectively. The dynamic yield strengths of LC and LU specimens were approximately equal. This was probably because the compressive strain and tensile strain were developed at the slope and center of rolled TWSPs, respectively. Therefore, much higher yield strength was attained for material at the center of rolled TWSPs compared with the LU specimen from the unrolled TWSPs due to strain stiffening, and a much lower yield strength for the material at the slope of rolled TWSPs due to Bauschinger effect. (2) The static, quasi-static, and dynamic tensile strengths of LS specimens were only 4%, 4%, and 2% lower than those of LU specimens, respectively, whereas those of LC specimens were 1%, 2%, and 3% lower, respectively, as presented in Figure 6b. The deviations of tensile strengths corresponding to different sampling locations of TWSPs were no greater than 5%. These results indicated that the cold rolling treatment had little effect on the tensile strength of the TWSP material, compared with its effect on the yield strength. A similar conclusion was also drawn in previous research regarding the effects of cold rolling treatment on the strength parameters of steel [27].

In terms of strain rate effects, the quasi-static yield strengths of LS, LC, and LU specimens were slightly higher than the static strengths of them, as depicted in Figure 6a. The quasi-static yield strengths of LS, LC, and LU specimens were no more than 10% higher than the static yield strengths of them. The dynamic yield strengths of LS, LC, and LU specimens clearly increased compared with the static strengths of them. The dynamic yield strengths of LS, LC, and LU specimens were 34%, 36%, and 42% higher than the static yield strengths of them, respectively. The effect of strain rate on tensile strength was similar to its effect on yield strength, as depicted in Figure 6b. Compared with the static tensile strengths, the quasi-static tensile strengths of LS, LC, and LU specimens were increased by 7%, 5%, and 7%, respectively, and the dynamic tensile strengths were increased by 20%, 16%, and 18%, respectively, as shown in Figure 6b.

#### 3.2.3. Analysis of Deformation Parameters

The total percentage extension at maximum force *A*_gt_ and percentage elongation after fracture *A* of the TWSP material are presented in Figure 7. The following observations could be made: (1) The cold rolling treatment had a significant influence on the static total percentage extension at maximum force, but its influence was negligible for the quasi-static and dynamic total percentage extension at maximum force. The static total percentage extensions at maximum force for LS (extracted from the slope of rolled TWSPs) and LC (extracted from the center of rolled TWSPs) specimens were 10% higher and 10% lower than that of LU specimens (extracted from unrolled flat TWSPs), respectively, as depicted in Figure 7a. (2) The percentage elongations after fracture of LC and LS specimens were higher than that of LU specimens under all strain rates, as indicated in Figure 7b, which indicated that the cold rolling treatment improved the deformability of the TWSP material. Compared with LU specimens, the static percentage elongations after fracture of LS and LC specimens increased by 13% and 35%, their quasi-static percentage elongations after fracture both increased by 8%, and their dynamic percentage elongations after fracture increased by 5% and 10%, respectively. (3) As the strain rates increased, the total percentage extensions at maximum force and the percentage elongations after fracture of the specimens decreased and increased, respectively, as shown in Figure 7. Compared with the static total percentage extensions at maximum force, the quasi-static total percentage extensions at maximum force of LS, LC, and LU specimens reduced by 27%, 6%, and 15%, respectively, but further reductions of 41%, 33%, and 35% were observed for their dynamic total percentage extensions at maximum force. Compared with the static percentage elongations after fracture, the quasi-static percentage elongations after fracture of LS, LC, and LU specimens increased by 23%, 2%, and 29%, respectively, and the dynamic percentage elongations after fracture increased by 26%, 10%, and 35%, respectively.

## 4. Dynamic Responses of the Rolled TWSPs Using the Mechanical Models in Section 3

### 4.1. Validation of the Numerical Model

To obtain more accurate dynamic responses of the rolled TWSP under impact loads, a numerical simulation method was verified based on the typical drop-hammer impact test conducted by Zeinoddini et al. [30] using explicit finite-element code LS-DYNA. As presented in Figure 8, specimen Pd1 with no axial pre-loading in Zeinoddini’s tests was used. The whole test system included a drop hammer, a fixed support, and a sliding support. The mass and impact speed of the drop hammer were 25.45 kg and 25.2 km/h, respectively. More detailed information about the test can be found in the reference [30].

Figure 8b depicts the numerical model of the drop-hammer impact test on a steel tube. The steel tube was simulated using four-noded, Belytschko–Tssy, thin-shell elements with three translational and three rotational degrees of freedom at each node [31]. The numbers of integration points for those elements were set to 1 and 3 in plane and thickness directions, respectively [32,33]. The drop hammer and sliding support at the ends of the steel tube were simulated using eight-noded constant stress solid elements with only three translational degrees of freedom at each node. The number of integration points was one at the center of volume for those solid elements. The material of the steel tube was simulated using the multilinear elastoplastic material model (*MAT_24), in which the strain rate effect was considered based on the Cowper–Symonds model, whose parameters *C* and *p* were set to 40 s^−1^ and 5, respectively [16]. An elastic material model (*MAT_01) was used for the drop hammer, steel plate, and sliding block. A rigid material (*MAT_20) was adopted for the rollers.

All six degrees of freedom for each node at the fixed end of the steel tube were fully constrained by the *BOUNDARY_SPC_SET keyword to fulfill the fixed support in the test, as depicted in Figure 8b. The sliding support of the steel tube was composed of a steel plate, a sliding block, and ten rollers, in which the steel plate was welded to the steel tube and sliding block. To ensure continuous transmission of force and displacement in the sliding support, the steel tube (shell elements) and the steel plate (solid elements) were connected using the keyword *CONTACT_TIED_SHELL_EDGE_TO_SOLID, which could couple the degrees of freedom of solid and shell elements. The steel plate and the sliding block (solid elements) were connected using the keyword *CONTACT_TIED_SURFACE_TO_SURFAC, which could couple the degrees of freedom of solid elements. The aforementioned method used for connection was also adopted by Naser and Degtyareva in simulating cold-formed steel beams subjected to shear [34]. The rotational degrees of freedom of the rollers along their axes were released, and the other translational degrees of freedom were restricted using *MAT 20, by which a sliding constraint was successfully achieved. An automatic surface-to-surface contact algorithm was applied between the drop hammer and the steel tube and the rollers and the sliding block, for which the static and dynamic friction coefficients were 0.74 and 0.57, respectively [35]. The monitored impact speed was directly assigned to the drop hammer using the keyword *INITIAL_VELOCITY. The mesh size was chosen as 5 mm in the study because limited improvement in predicting the displacement and force (less than 0.5%) was achieved when using a finer mesh.

Figure 9 presents a comparison between the numerical and experimental results for time history curves and deformation pattern of the steel tube. The numerical results obtained by Zeinoddini et al. [36] are also plotted in Figure 9a,b. The deviation between the numerical and experimental results for maximum impact force was reduced to 0% (simulated in this study) from 14% (simulated by Zeinoddini et al.). The deviation between the numerical and experimental results for maximum axial displacement was reduced to 17% (simulated in this study) from 25% (simulated by Zeinoddini et al.). The numerical model was able to capture the impact force and axial displacement of the steel tube with reasonable accuracy. The dent depth in simulation was 7% lower than that in experiment, as shown in Figure 9c, which indicated that the numerical model accurately captures the deformation pattern of the steel tube under impact load. Therefore, the numerical simulation method could be used to simulate the dynamic responses of rolled TWSPs under the impact load shown in the following sections.

### 4.2. Numerical Model of the Drop Hammer Test on the Rolled TWSP

Based on the numerical simulation method verified by the drop hammer test on the steel tube in Section 4.1, a numerical model was established to study the dynamic responses of the rolled TWSP without considering the influences of geometric imperfections, as depicted in Figure 10. The rolled TWSP was 4.32 m in length, and its cross-section is illustrated in Figure 2. The multi-linear elastoplastic material model (*MAT_24) was also adopted for the rolled TWSP. Rather than using the Cowper–Symonds model, the material’s strain rate effect was considered directly by adopting the equivalent stress–equivalent plastic strain curves under different strain rates. Through trial analysis, the impact speed of the 25.45 kg drop hammer was set to 20 km/h to ensure that the maximum strain rate of the TWSP material was less than 50 s^−1^. The magnitude and direction of impact speed were directly assigned to the drop hammer using the keyword *INITIAL_VELOCITY. The desired mass of the drop hammer was achieved by adjusting density considering its fixed volume in the numerical simulation. Other parameter settings, such as boundary conditions; material models of the drop hammer, steel plate, sliding block, and rollers; and element types, were the same as in the numerical model described in Section 4.1.

Four numerical models (labeled Model LS, Model LC, Model LU, and Model RS) with different mechanical models for the rolled TWSP were studied, as shown in Figure 10a–d. For Models LS, LC, and LU, the material model of rolled TWSP used the equivalent stress–equivalent plastic strain curves measured by LS, LC, and LU specimens, respectively. For Model RS, the equivalent stress–equivalent plastic strain curves of LS specimens were used for material at the cross-sectional slope of the rolled TWSP, whereas those of LC specimens were used for the material at cross-sectional center of the rolled TWSP to better reflect the varying material properties along the cross-section of rolled TWSP. Model RS was taken as the contrast specimen because it could represent the structural dynamic responses of the rolled TWSP in real situations.

### 4.3. Dynamic Responses of the Rolled TWSP

Figure 11 presents the time history curves of impact force impulse and lateral displacement of the rolled TWSP subjected to the impact of a drop hammer. The residual values of impact force impulse in Models LS, LC, and LU were 17%, 8%, and 10% lower than that in Model RS, respectively. The maximum lateral displacements in Models LS and LU were 9% and 5% higher than that in Model RS, respectively, while that in Model LC was only 2% lower than that in Model RS. The residual lateral displacement in Model LS was 10% higher than that in Model RS, whereas those in Models LC and LU were 6% and 13% lower than that in Model RS, respectively. These results indicate that Model LC and Model RS had approximate simulation results, which included the impact force impulse and lateral displacement of the rolled TWSP. Therefore, the cross-sectional center of the rolled TWSP is recommended as the representative sampling location for the rolled TWSP material in this study. Furthermore, although relatively large differences were exhibited among the mechanical properties of the specimens extracted from different sampling locations (as discussed in Section 3), the dynamic responses of the rolled TWSP under impact load were less different when the mechanical properties of material from the above three sampling locations were used in numerical simulation. This was probably because the plastic deformation did not thoroughly develop under the impact speed of 20 km/h.

## 5. Mechanical Model of the Rolled TWSP under Low and Medium Strain Rates

### 5.1. Test Specimens and Results

Based on the conclusions in Section 4.3, the cross-sectional center of the rolled TWSP is recommended as the representative sampling location. Therefore, additional uniaxial tensile tests (labeled LC-D10, LC-D100, LC-D200) with strain rates of 10, 100, and 200 s^−1^ were conducted on LC specimens to calibrate the mechanical model so that it could reasonably reflect the strain effects on the rolled TWSP material under impact load. The test setup and measurement devices in this part were identical to the dynamic tests discussed in Section 2.

Figure 12 presents the relationship between the measured engineering stress and engineering strain under strain rates ranging from 0.00025 to 200 s^−1^. The strength and deformation parameters at various strain rates are listed in Table 4. Figure 12 and Table 4 indicate that as the strain rates increased, the yield strength, tensile strength, and percentage elongation after fracture also increased, but the total percentage extension at maximum force decreased. The increases in the yield strength and tensile strength were due to enhancement of the activation energy of dislocations to bypass obstacles under higher strain rates [37,38]. The percentage elongation after fracture was increased because the ductility of the material was enhanced under dynamic loading and inertia [39].

### 5.2. Calibration of Cowper–Symonds Model Parameters

The Cowper–Symonds model is a widely used mechanical model in finite element software. A dynamic increase factor (*DIF*) is given in the model to reflect the material strain rate’s effect on strength, as formulated in Equation (5), enabling the dynamic true stress *σ*_true__, d_ to be obtained by correcting the static true stress *σ*_true__, s_ [16].
(5)$${DIF\;=\sigma }_{\mathrm{true},d}{/\sigma }_{\mathrm{true},s}=\;1\;+(\dot{\varepsilon }{/C)}^{1/p}$$
where $\dot{\varepsilon }$ is the strain rate; and *C* and *p* are the model parameters related to strain rates.

Figure 13 presents the *DIF* calculated from yield strength and tensile strength using Equation (1). The *DIF* calculated by the Cowper–Symonds model using parameters of *C* = 40 s^−1^ and *p* = 5 is also plotted in Figure 13. It indicates that the model with parameters of *C* = 40 s^−1^ and *p* = 5 suggested by Symonds [40] significantly overestimated the *DIF* of the rolled TWSP material for strain rates ranging from 0.00025 to 200 s^−1^. Therefore, an additional analysis was performed to quantify the influence of strain rate on the strength of the rolled TWSP material. The relationship curves between the *DIF* and the strain rate were obtained by the methods of fitting the Cowper–Symonds model based on yield strength and tensile strength [41]. The values of *C* and *p* were 2635 s^−1^ and 3.8, respectively, when the Cowper–Symonds model parameters were fitted by yield strength, but changed to 6993 s^−1^ and 2.0, respectively, when they were fitted by yield strength. The standard errors of estimate (SEE) were 0.001 and 0.0004 for the calibrated Cowper–Symonds models fitted by yield strength and tensile strength, respectively, indicating the goodness of the fitting method.

### 5.3. Validation of the Calibrated Cowper–Symonds Models

Figure 14a,c compares the experimental true stress–strain curves of the rolled TWSP material with the calculated true stress–strain curves using the model parameters calibrated in Section 5.2 and suggested by Symonds. To evaluate the accuracies of different models, the SEE of Cowper–Symonds models with different parameters of *C* and *p* are listed in Table 5. The smaller the SEE, the closer the stress–strain curves calculated by Cowper–Symonds models to the experimental stress–strain curves. Figure 14 and Table 5 indicate that the Cowper–Symonds model with parameters of *C* = 40 s^−1^ and *p* = 5 significantly overestimated the strain rate effect of the rolled TWSP material with a SEE of 256.02. The Cowper–Symonds model with parameters of *C* = 2635 s^−1^ and *p* = 3.8 fitted by tensile strength also overestimated the strain rate effect of the rolled TWSP material with a SEE of 54.32. The Cowper–Symonds model with parameters of *C* = 6993 s^−1^ and *p* = 2.0 underestimated the strain effect of the rolled TWSP with a SEE of 32.76. It needs to be noted that although the Cowper–Symonds model parameters fitted by yield strength and tensile strength in Section 5.2 had high fitting goodness for *DIF*, the calculated stress–strain curves using above model parameters do not correlate well with the experimental stress–strain curves. This was because the strain and strain rate had coupled effects on the *DIF*, but the values of *C* and *p* were fitted only based on the critical point (e.g., yield strength or tensile strength), which only represented the effects of strain rate on *DIF* [25].

Based on above analysis, the Cowper–Symonds model with parameters of *C* = 4814 s^−1^ and *p* = 2.9, which are the average values of model parameters fitted by yield strength and tensile strength, was proposed for the rolled TWSP material. Figure 14d presents the comparison between the calculated stress–strain curves using the Cowper–Symonds model with parameters of *C* = 4814 s^−1^ and *p* = 2.9 and experimental stress–strain curves of the rolled TWSP. This indicates that the Cowper–Symonds model with parameters of *C* = 4814 s^−1^ and *p* = 2.9 provided a reasonable prediction for the true stress–strain curves of the rolled TWSP material with a SEE of 22.54.

## 6. Dynamic Responses of the Rolled TWSP Using Mechanical Models in Section 5

Five numerical models (labeled Model-A, Model-B, Model-C, Model-D, and Model-E) were used to study the influences of strain rate effects on the dynamic responses of the rolled TWSPs. Model-A to Model-D were identical to the numerical model described in Section 4.2, except that Cowper–Symonds models were used for the rolled TWSPs to consider strain rate effects. For Model-A, *C* and *p* in Cowper–Symonds model were 40 s^−1^ and 5, respectively, as recommended by Symonds. For Model-B, *C* and *p* were 2635 s^−1^ and 3.8, respectively, as calibrated by tensile strength. For Model-C, C and *p* were 6993 s^−1^ and 2.0, respectively, as calibrated by yield strength. For Model D, C and *p* were 4814 s^−1^ and 2.9, respectively, as proposed in Section 5.3. For Model-E, the experimental equivalent stress–equivalent plastic strain curves under different strain rates were directly used for the rolled TWSP to consider strain rate effects. The rolled TWSP in Model-E was taken as the contrast specimen because it could represent the dynamic responses of the rolled TWSP in W-beam guardrails in real situations. Three drop hammer speeds of 20, 60, and 100 km/h were considered in the study, which could confirm that the maximum strain rate of the rolled TWSP did not exceed 200 s^−1^ by trial analysis.

Figure 15 presents the dynamic responses of the rolled TWSP using different Cowper–Symonds models. It indicates that the five different Cowper–Symonds models had a limited influence on the time history curves of contact force impulse of the rolled TWSP, but a larger influence on the time history curves of the deformation of it. The lateral displacement of the rolled TWSP in Model-C was highest, whereas that in model-A was lowest. The lateral displacement of the rolled TWSP in Model-E was higher and lower than those in Model-B and Model-D, respectively. For example, when the drop hammer impacted the rolled TWSP at 100 km/h, the residual lateral displacements of rolled TWSP in Model-C, -D, -E, -B, and -A were 219, 209, 204, 194, and 139 mm, respectively.

Comparisons of the maximum lateral displacement and residual lateral displacement for Model-A to Model-E are shown in Figure 16. The following observations could be made: (1) The maximum lateral displacement of the rolled TWSP in Model-A decreased by 4%, 19%, and 21% for the three impact speeds of 20, 60, and 100 km/h, respectively, compared with Model-E, indicating the percentage decreases increased as impact velocity increased. This was mainly because the strain rate of the rolled TWSP material increased as the impact velocity increased, and the larger the strain rate, the more severe the overestimation of the strain rate effect on the rolled TWSP material for Model-A (see Figure 16a). The maximum lateral displacements of the other three models (Model-B to Model-D) were close to that of Model-E, especially Model-D. The percentage variations of maximum lateral displacement between Model-D and Model-E were 0%, 1%, and 1% at the three impact speeds, respectively. (2) Compared with Model-E, the percentage decreases in the residual lateral displacement of rolled TWSP in Model-A were 70%, 37%, and 32% at the three impact speeds of 20, 60, and 100 km/h; the percentage decreases in Model-B were 14%, 8% and 5%; the percentage increases in Model-C were 16%, 8%, and 7%; and the percentage variations in Model-D were −8%, 8%, and 7%, respectively, as shown in Figure 16b. (3). The maximum lateral displacement and residual lateral displacement of the rolled TSWP were predicted well using the Cowper–Symonds model with parameters of “*C* = 2635 s^−1^ and *p* = 3.8”; “*C* = 6993 s^−1^ and *p* = 2.0”; or “*C* = 4814 s^−1^ and *p* = 2.9.” Furthermore, the lateral displacements of the rolled TSWP were predicted most reasonably using the Cowper–Symonds model with parameters *C* = 4814 s^−1^ and *p* = 2.9, but were significantly overestimated using the authoritative Cowper–Symonds model with parameters of *C* = 40 s^−1^ and *p* = 5. It could be concluded that the Cowper–Symonds model with calibrated parameters of *C* = 4814 s^−1^ and *p* = 2.9 provided not only a reasonable prediction for the true stress-stain curves of the rolled TWSP material in Section 5.3, but also accurate dynamic responses (i.e., time history curves of contact force impulse and lateral displacement) of the rolled TWSP under impact load. Therefore, Cowper–Symonds model with calibrated parameters of *C* = 4814 s^−1^ and *p* = 2.9 is recommended for considering the strain rate effects on the rolled TWSP material in the low and medium strain-rate regions.

## 7. Conclusions

To establish an accurate mechanical model of the rolled TWSP material under low and medium strain rates, static, quasi-static, and dynamic uniaxial tensile tests were conducted on specimens extracted from flat TWSPs and the cross-sectional centers and slopes of rolled TWSPs. The main conclusions are as follows:(1)The cold rolling treatment had little effect on the tensile strength of the TWSP material, compared with its effect on the yield strength. The static ($\dot{\varepsilon }$= 0.00025 s^−1^), quasi-static ($\dot{\varepsilon }$ = 0.01 s^−1^), and dynamic ($\dot{\varepsilon }$ = 50 s^−1^) yield strengths of material at the cross-sectional slope of the rolled TSWP were 10%, 7%, and 15% lower than those of flat TWSP material, respectively. The static, quasi-static, and dynamic yield strengths of material at the cross-sectional center of the rolled TWSP were no more than 6% higher than those of the flat TWSP. The deviations in tensile strengths among material from different sampling locations of the TWSP were no greater than 5%. The cold rolling treatment also improved the deformability of the TWSP material.(2)The strain rate had a stronger effect on the yield strength than the tensile strength of the rolled TWSP material. The quasi-static and dynamic yield strength of the rolled TWSP material were no more than 10% and 36% higher than the static yield strength, respectively. The quasi-static and dynamic tensile strength were increased by no more than 7% and 20% compared with the static tensile strength, respectively. The total percentage extensions at maximum force and the percentage elongations after fracture of the rolled TWSP material decreased and increased as strain rates increased, respectively.(3)The cross-sectional center of rolled TWSPs is recommended as the representative sampling location for uniaxial tensile tests of rolled TWSP material because the dynamic responses of rolled TWSPs were reasonably captured when the stress–strain curves measured by the specimen extracted from cross-sectional center of rolled TWSP were used in simulation.(4)The strain rate effects of rolled TWSP material were significantly overestimated by the Cowper–Symonds model with parameters of *C* = 40 s^−1^ and *p* = 5 for strain rates ranging from 0.00025 to 200 s^−1^, but they were predicted well by the Cowper–Symonds model with parameters *C* = 4814 s^−1^ and *p* = 2.9 recommended in this study.

## Figures and Tables

**Figure 1 materials-15-06504-f001:**
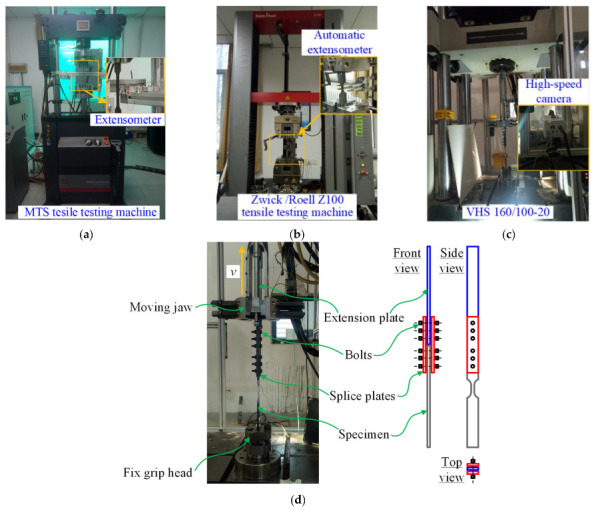
Test setup and measurement devices. (**a**) Static test. (**b**) Quasi-static test. (**c**) Dynamic test. (**d**) Clamping devices in the dynamic test.

**Figure 2 materials-15-06504-f002:**
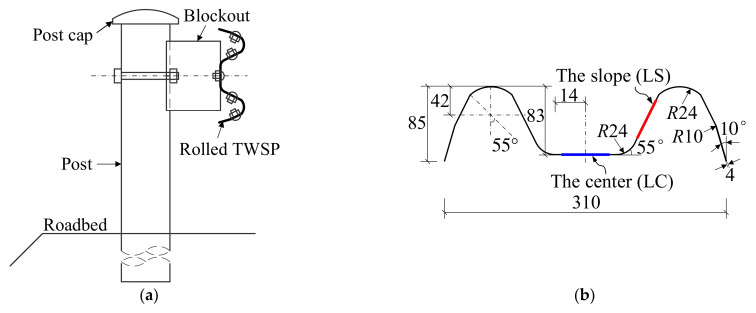
Rolled TWSP. (**a**) Position in W-beam guardrail. (**b**) Cross-section (unit: mm).

**Figure 3 materials-15-06504-f003:**
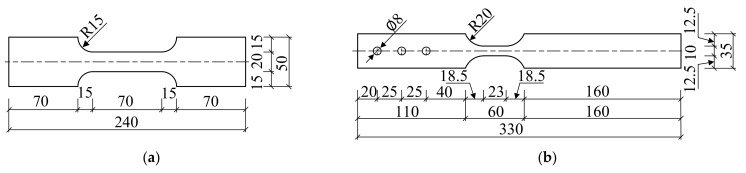
Geometric dimensions of test specimens (unit: mm). (**a**) Static tests. (**b**) Quasi-static and dynamic tests.

**Figure 4 materials-15-06504-f004:**
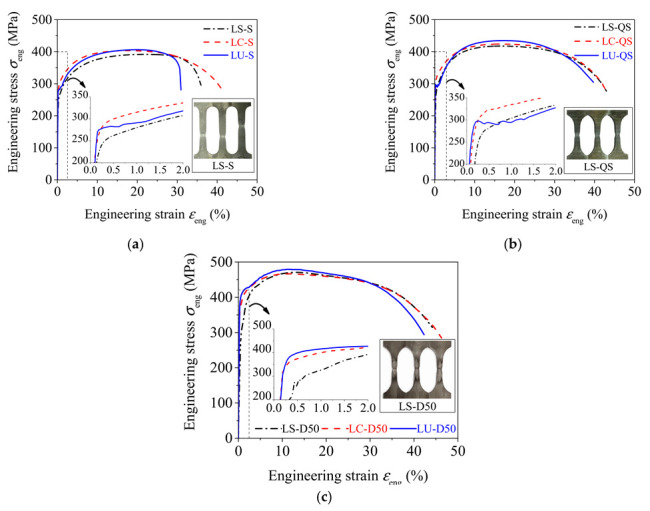
Engineering stress–strain curves. (**a**) Static test, $\dot{\varepsilon }\;$ = 0.00025 s^−1^. (**b**) Quasi-static test, $\dot{\varepsilon }\;$ = 0.01 s^−1^. (**c**) Dynamic test, $\dot{\varepsilon }\;$ = 50 s^−1^.

**Figure 5 materials-15-06504-f005:**
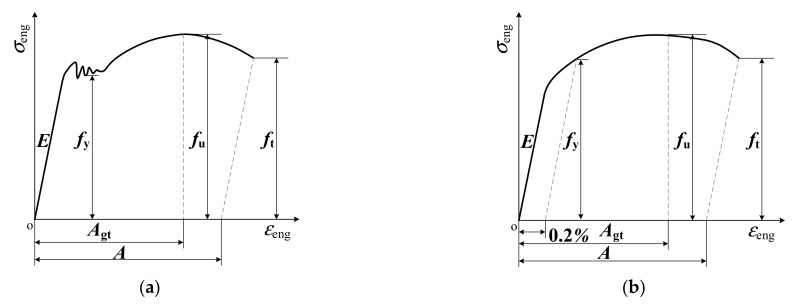
Definitions of strength and deformation parameters for the TWSPs. (**a**) With yield plateau. (**b**) Without yield plateau.

**Figure 6 materials-15-06504-f006:**
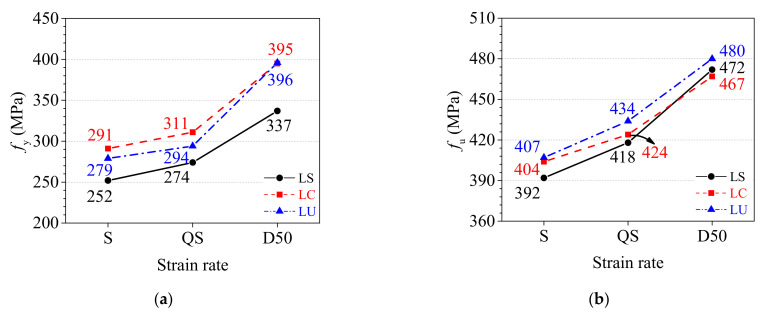
Strength parameters under three strain rates. (**a**) Yield strength. (**b**) Tensile strength.

**Figure 7 materials-15-06504-f007:**
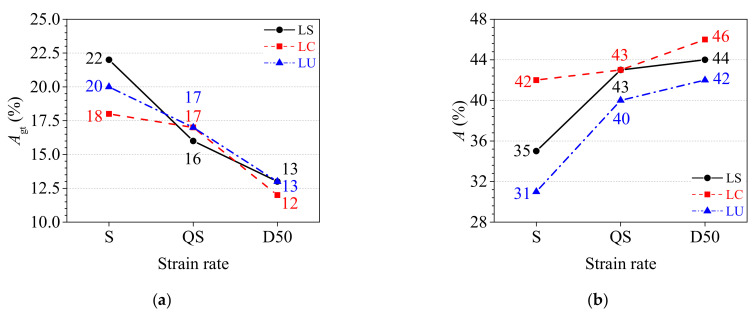
Deformation parameters under three strain rates. (**a**) Percentage total extension at maximum force. (**b**) Percentage elongation after fracture.

**Figure 8 materials-15-06504-f008:**
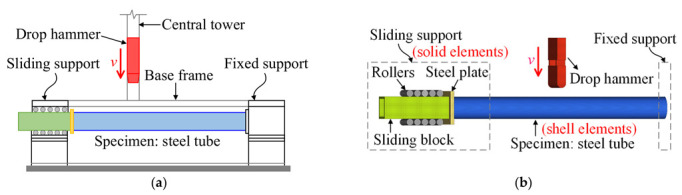
Numerical simulation of the drop hammer test on a steel tube. (**a**) Test setup. Reprinted with permission from Ref. [30]. 2002, Elsevier (**b**) Numerical model.

**Figure 9 materials-15-06504-f009:**
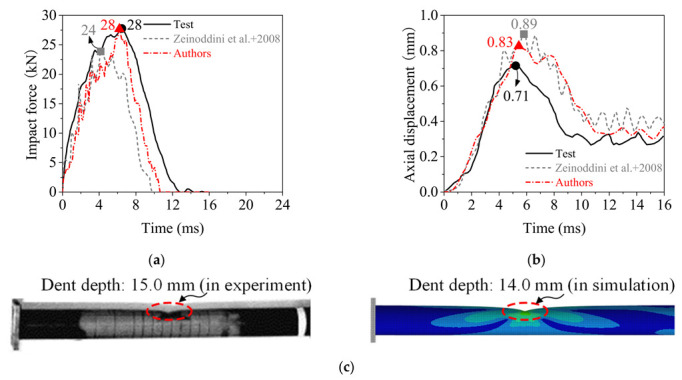
Comparison of numerical and experimental results. Reprinted with permission from Refs. [30,36]. 2008, Elsevier. (**a**) Impact force. (**b**) Axial displacement. (**c**) Deformation pattern.

**Figure 10 materials-15-06504-f010:**
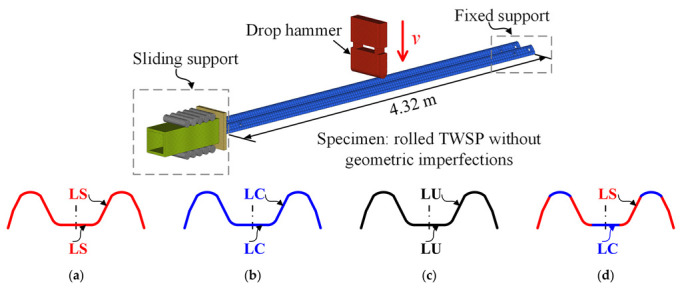
Numerical model of the drop hammer test on the rolled TWSP. (**a**) Model LS. (**b**) Model LC. (**c**) Model LU. (**d**) Model RS.

**Figure 11 materials-15-06504-f011:**
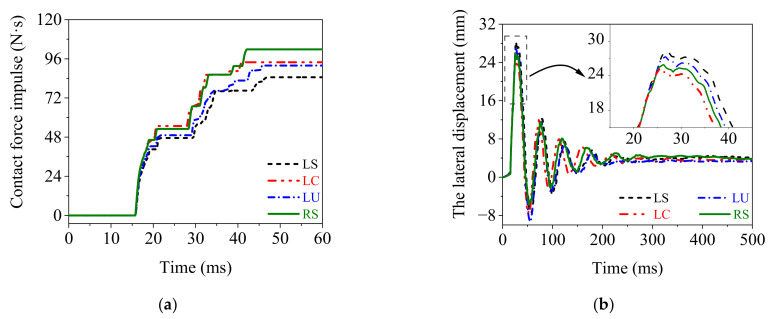
Dynamic responses of the rolled TWSP under impact load. (**a**) Impact force impulse. (**b**) Lateral displacement.

**Figure 12 materials-15-06504-f012:**
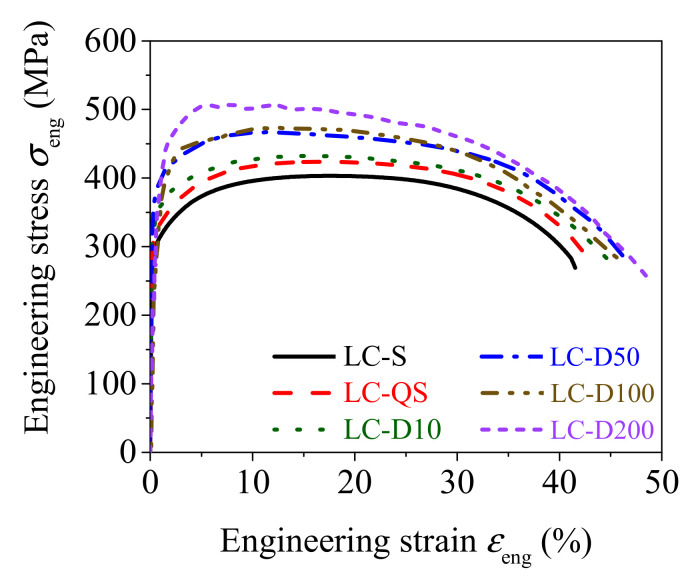
Engineering stress–strain curves of LC specimens ($\dot{\varepsilon \;}$ = 0.00025 to 200 s^−1^).

**Figure 13 materials-15-06504-f013:**
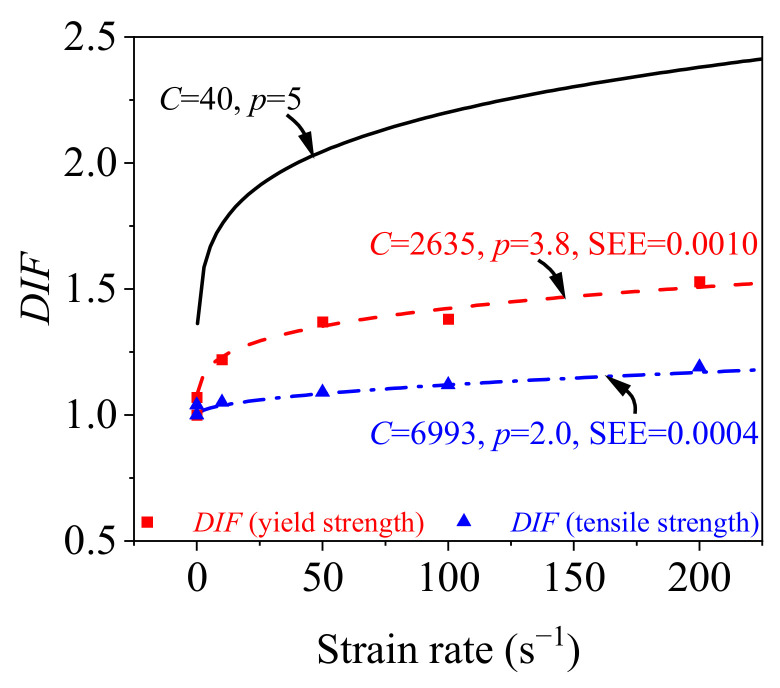
Relationship between DIF and strain rate.

**Figure 14 materials-15-06504-f014:**
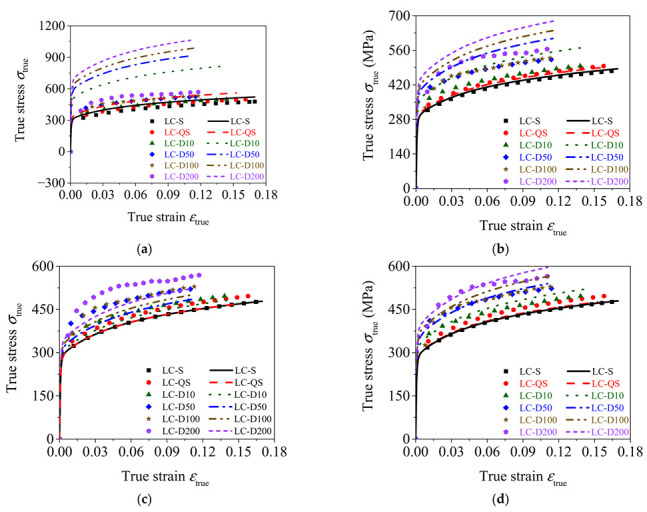
Comparison between the calculated and experimental stress–strain curves. (**a**) *C* = 40 s^−1^, *p* = 5. (**b**) *C* = 2635 s^−1^, *p* = 3.8. (**c**) *C* = 6993 s^−1^, *p* = 2.0. (**d**) *C* = 4814 s^−1^, *p* = 2.9.

**Figure 15 materials-15-06504-f015:**
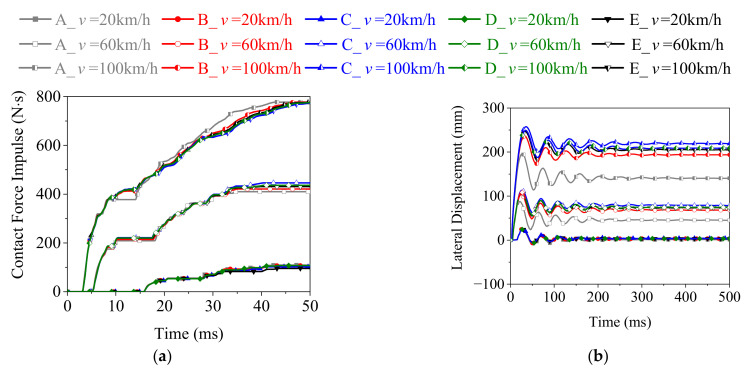
Dynamic responses of rolled TWSP using different Cowper–Symonds models. (**a**) Contact force impulse. (**b**) Lateral displacement.

**Figure 16 materials-15-06504-f016:**
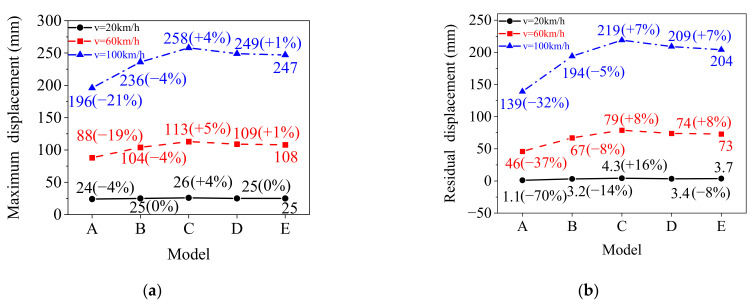
Maximum and residual lateral displacement of the rolled TWSP. (**a**) Maximum lateral displacement. (**b**) Residual lateral displacement. Note: values in parentheses refer to percentage increases and decreases with respect to model-E.

**Table 1 materials-15-06504-t001:** Load capacities and loading velocities of the three testing machines.

Tensile Testing Machine	Test Pattern	Load Capacity (kN)	loading Velocity (mm/min)
MTS	static	100	0.001–508
Zwick/Roell Z100	quasi-static	100	0.0005–1500
Instron VHS 160/100-20	dynamic	100	≤1,200,000

**Table 2 materials-15-06504-t002:** Test specimens.

Strain Rate (s^−1^)	Static $(S,\;\dot{\varepsilon }$ = 0.00025 s^−1^)	Quasi-Static $(\mathrm{QS},\;\dot{\varepsilon }$ = 0.01 s^−1^)	Dynamic $(D50,\;\dot{\varepsilon }$ = 0.50 s^−1^)
Slope of rolled TWSP (LS)	LS-S	LS-QS	LS-D50
Center of rolled TWSP (LC)	LC-S	LC-QS	LC-D50
Unrolled flat TWSP (LU)	LU-S	LU-QS	LU-D50

**Table 3 materials-15-06504-t003:** Mechanical parameters of LS, LC, and LU specimens ($\dot{\varepsilon }$ = 0.00025, 0.01, and 50 s^−1^).

Strain Rates (s^−1^)		S ( $\dot{\varepsilon }$ = 0.00025)			QS ( $\dot{\varepsilon }$ = 0.01)			D50 ( $\dot{\varepsilon }$ = 0.50)		
Sample Locations		LS	LC	LU	LS	LC	LU	LS	LC	LU
Strength parameters	*f*_y_ (MPa)	252	291	279	274	311	294	337	395	396
	*f*_u_ (Mpa)	392	404	407	418	424	434	472	467	480
	*f*_t_ (Mpa)	297	272	300	277	282	305	313	283	297
Deformation parameters	*E* (Gpa)	188	243	204	-	-	-	-	-	-
	*A*_gt_ (%)	22	18	20	16	17	17	13	12	13
	*A* (%)	35	42	31	43	43	40	44	46	42

**Table 4 materials-15-06504-t004:** Strength and deformation parameters of LC specimens ($\dot{\varepsilon \;}$ = 0.00025, 0.01, and 50 s^−1^).

Labels of Specimens (s^−1^)		Tests in Section 3			Additional Tests		
		LC-S $(\dot{\varepsilon }$ = 0.00025)	LC-QS $(\dot{\varepsilon }$ = 0.01)	LC-D50 $(\dot{\varepsilon }$ = 0.50)	LC-D10 $(\dot{\varepsilon }$ = 10)	LC-D100 $(\dot{\varepsilon }$ = 100)	LC-D200 $(\dot{\varepsilon }$ = 200)
Strength parameters	*f*_y_ (MPa)	291	311	395	353	398	438
	*f*_u_ (Mpa)	404	424	467	433	475	509
	*f*_t_ (Mpa)	272	282	283	266	248	258
Deformation parameters	*A*_gt_ (%)	18	17	12	15	12	12
	*A* (%)	42	43	46	45	46	48

**Table 5 materials-15-06504-t005:** Fitting goodness of Cowper–Symonds model evaluated by SEE.

Labels of Specimens (s^−1^)	*C* = 40 s^−1^, *p* = 5	*C* = 2635 s^−1^, *p* = 3.8	*C* = 6993 s^−1^, *p* = 2.0	*C* = 4814 s^−1^, *p* = 2.9
LC-S $(\dot{\varepsilon }$ = 0.00025)	38.32	5.97	0.08	1.30
LC-QS $(\dot{\varepsilon }$ = 0.01)	57.20	6.43	21.23	17.21
LC-D10 $(\dot{\varepsilon }$ = 10)	275.24	59.68	21.38	15.54
LC-D50 $(\dot{\varepsilon }$ = 50)	333.90	62.11	46.77	13.90
LC-D100 $(\dot{\varepsilon }$ = 100)	402.76	100.60	45.90	48.12
LC-D200 $(\dot{\varepsilon }$ = 200)	428.69	91.12	61.22	39.19
Average value	256.02	54.32	32.76	22.54

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
