# Peer review of "Dynamic Mechanical Properties of Rolled Thin-Walled Steel Plates (TWSPs) Used for W-Beam Guardrails under Low and Medium Strain Rates"

_materials, 2022, doi:10.3390/ma15196504_

Round 1
Reviewer 1 Report
An interesting paper on the dynamics mechanical properties of TWSPs. A few items arise during the review process:
1. How does the location of the extracted coupon affect the measured properties?
2. Please cite the used material testing stadnard.
3. Please explain why "The static percentage total extensions at maximum force for LS and LC specimens were 10% higher and 10% lower than that of LU specimens"
4. The details on the FE models are quite superficial and light. Please ensure that your model is properly described to enable interested researchers from extending and replicating your work. Special attention should be paid to specifics such as element type (DOFs), convergence criteria and performance metrics. The authors may refer to the work of https://doi.org/10.1016/j.tws.2018.12.030 for details on FE models.
5. Did the authors consider geometric imperfections. How about the effect of residual stresses?
Author Response
The authors thank the reviewer very much for the comments and appreciate the time and effort for the thorough review. The point-to-point reply to those specific comments are given in the attached file.

Reviewer 2 Report
This study deals with the dynamic mechanical properties of rolled tin-walled steel plates used for W-beam guardrails under low and medium strain rates. The content is suitable to publish on Materials. The reviewer has some comments as follows:
1) The novelty and contribution of the study must be clarified in section 1.
2) The temperature room in the experiment needs to be stated in a specific number (line 96).
3) The relationship between test specimens (Figure 3) and rolled TWSP’s cross section (Figure 2b) should be explained.
4) How to model the 25.45 kg drop hammer and 20 km/h impact speed in the numerical simulation? (in section 4.2, line 243).
5) The quality of Figures needs to be improved.
6) The English needs to be polished.
Author Response

(The authors gave the same response as above.)

Reviewer 3 Report
In this study, static, quasi-static, and dynamic uniaxial tensile tests with strain rates of 0.00025 s-1, 0.01 s-1, and 50 s-1, respectively, were firstly conducted on specimens extracted from a flat TWSP (without cold rolling treatment) and the cross-sectional center and slope of a rolled TWSP (with cold rolling treatment). The influences of cold rolling treatment on the mechanical properties of TWSPs were investigated. A representative sampling location is recommended based on the test results and numerical analysis of rolled TWSPs. Based on the representative sampling location, additional dynamic uniaxial tensile tests with wider strain rates of 10 s-1, 100 s-1, and 200 s-1 were conducted to study the influence of strain rates on the mechanical properties of the rolled TWSP material. Finally, the parameters of the Cowper-Symonds model were calibrated to ensure the Cowper-Symonds model could reasonably reflect the dynamic mechanical properties of the rolled TWSPs. The introduction clearly sets out the problem under investigation. The methodology is adequate. Model formulation is good presented. Model Setup and Validation are described. Several minor issues need to be solved before further consideration.
- results and discussion and conclusion parts are inadequate according to citation and analyze in detail. There should be the importance of the study in detail, comparison results with other approaches in literature, the success of the prediction and computational results.
- the accuracies of the related devices should be listed in the table.
- the range of a article of 34 pages is long enough for me, it could be 2 articles
- the Abstact part needs to be improved. The authors needs to describe the main results in the abstract.
The paper is acceptable for publication with minor revisions.
Author Response

(The authors gave the same response as above.)

Round 2
Reviewer 1 Report
.